# Spotted Fever Group *Rickettsia* spp. Diversity in Ticks and the First Report of *Rickettsia hoogstraalii* in Romania

**DOI:** 10.3390/vetsci9070343

**Published:** 2022-07-08

**Authors:** Talida Ivan, Ioana Adriana Matei, Cristiana Ștefania Novac, Zsuzsa Kalmár, Silvia-Diana Borșan, Luciana-Cătălina Panait, Călin Mircea Gherman, Angela Monica Ionică, Ionel Papuc, Andrei Daniel Mihalca

**Affiliations:** 1Department of Semiology, Faculty of Veterinary Medicine, University of Agricultural Sciences and Veterinary Medicine of Cluj-Napoca, 400372 Cluj-Napoca, Romania; talida.holmic@usamvcluj.ro (T.I.); ionel.papuc@usamvcluj.ro (I.P.); 2Department of Microbiology, Immunology and Epidemiology, Faculty of Veterinary Medicine, University of Agricultural Sciences and Veterinary Medicine of Cluj-Napoca, 400372 Cluj-Napoca, Romania; cristiana.novac@usamvcluj.ro (C.Ș.N.); zsuzsa.kalmar@usamvcluj.ro (Z.K.); 3Department of Infectious Diseases, Iuliu Hațieganu University of Medicine and Pharmacy Cluj-Napoca, 400347 Cluj-Napoca, Romania; 4Clinical Hospital of Infectious Diseases of Cluj-Napoca, 400003 Cluj-Napoca, Romania; angela.ionica@usamvcluj.ro; 5Department of Parasitology and Parasitic Diseases, Faculty of Veterinary Medicine, University of Agricultural Sciences and Veterinary Medicine of Cluj-Napoca, 400372 Cluj-Napoca, Romania; silvia.borsan@usamvcluj.ro (S.-D.B.); luciana.rus@usamvcluj.ro (L.-C.P.); calin.gherman@usamvcluj.ro (C.M.G.); amihalca@usamvcluj.ro (A.D.M.)

**Keywords:** *Rickettsia hoogstraalii*, Romania, SFG *Rickettsia* spp. diversity, ticks

## Abstract

**Simple Summary:**

Ticks are important parasites that feed on the blood of various host species, representing the most important arthropods transmitting diseases in Europe. Continuous changes in both tick distribution and abundance are related to multiple factors, including climate change. These changes have strong implications for both animal and human health; therefore, continuous surveillance of tickborne diseases is required for an appropriate evaluation of the potential risks faced by animals and humans in a given area. The spotted fever group Rickettsia comprises a large number of zoonotic agents with an increasing importance recognized in the last 30 years. The aim of this study was to evaluate these bacteria in ticks in Romania. Five Rickettsia species were identified in different tick species, with new pathogen–tick associations reported. *Rickettsia hoogstraalii*, one member of this group, was detected for the first time in Romania and in *Rhipicephalus rossicus* ticks. This species was first described in 2006 in Croatia, and its pathogenicity is not well known. In addition, the detection of *R. raoultii* and *R. monacensis* in unfed larvae of *Haemaphysalis punctata* reinforce the hypothesis of transmission of *Rickettsia* from female ticks to larvae; therefore the bite of larvae could pose a health risk.

**Abstract:**

Tickborne bacterial pathogens have been described worldwide as risk factors for both animal and human health. Spotted fevers caused by Rickettsiae may cause non-specific symptoms, which make clinical diagnosis difficult. The aim of the current study was to evaluate and review the diversity of SFG Rickettsiae in ticks collected in 41 counties in Romania. A total of 2028 questing and engorged ticks collected in Romania belonging to five species were tested by PCR amplification of *Rickettsia* spp. gltA and 17-D gene fragments: *Ixodes ricinus* (*n* = 1128), *Dermacentor marginatus* (*n* = 507), *D. reticulatus* (*n* = 165), *Rhipicephalus rossicus* (*n* = 128) and *Haemaphysalis punctata* (*n* = 100). Five *Rickettsia* species were identified following DNA sequence analysis: *R. helvetica*, *R. monacensis*, *R. slovaca*, *R. raoultii*, and *R. hoogstraalii.* The most common species detected was *R. monacensis*. Moreover, *R. hoogstraalii* was detected for the first time in Romania and in *R. rossicus* ticks. The detection of *R. raoultii* and *R. monacensis* in questing larvae of *Hae. punctata* suggests the possible transovarial transmission of these *Rickettsia* species in ticks. The detection of *R. hoogstraalii* for the first time in Romania increases the reported SFG *Rickettsia* diversity in the country.

## 1. Introduction

Among the most common vectors of disease in Europe, ticks are important hematophagous ectoparasites with a worldwide distribution and the ability to transmit a wide variety of pathogens [1]. The population and community structure, as well as their abundance, are related to the geographical origin and are influenced by various abiotic and biotic factors [2,3,4]. The continuous changes in both tick distribution and abundance are related to multiple factors, such as climate change, habitat alterations, biodiversity loss, and globalisation [5,6]. In addition, land use has direct effects on the ecosystem through cross interactions between pathogens, hosts, and reservoirs, thus enabling the exposure of hosts to potential pathogens [7]. In addition, some tick species (e.g., *Ixodes ricinus*) possess a broad ecological plasticity, with increased capacity to exploit anthropic landscapes, which has led to the emergence and re-emergence of several tickborne diseases, with strong implications for both animal and human health [8,9]. Considering the ongoing changes in the above-mentioned factors and their effects on both ticks and their associated pathogens, continuous surveillance of tickborne diseases is required for an appropriate evaluation of the potential risks faced by animals and humans in a given area.

The spotted fever group (SFG) of the *Rickettsia* genus comprises a large number of zoonotic agents. The importance of the recognized tick-associated rickettsial pathogens has increased in the last 30 years. Moreover, the pathogenicity for humans of several species has been continuously described, and novel *Rickettsia* species of undetermined pathogenicity have been detected in ticks around the world [10,11]. *Rickettsia* spp. are Gram-negative bacteria with intracellular development [12] belonging to the Class α-proteobacteria, Order Rickettsiales, Family Rickettsiaceae [13]. *Rickettsia* are transmitted mainly through arthropod vectors, with an important number being transmitted by ticks [14]. In Europe, the majority of *Rickettsia* infections are tickborne [15]. *Rickettsia* species of medical concern detected in *I. ricinus*, one of the most widespread tick species in Europe [16], include *R. helvetica* [17], *R. monacensis*, *R. raoultii* [18,19], *R. slovaca*, and *R. sibirica mongolitimonae* [19]. In addition, agents with unknown pathogenicity, such as: *Candidatus* “R. mendelii” [20], *R. bellii*, and *Rickettsia* endosymbiont of *Lasioglossum semilucens* bee [21] have been reported in this tick species in countries such as the Czech Republic [20], Germany [14], Poland [21], and Sweden [19].

Other ticks, such as *Rhipicephalus sanguineus* [14], *R. pumilio*, *R. turanicus* [17], *Dermacentor* spp. [14], and *Hyalomma marginatum* [22], have been shown or suggested to be involved in the circulation and transmission of other *Rickettsia* spp., such as *R. conorii conorii* [14], *R. raoultii*, *R. slovaca* [14], *R. monacensis*, *R. helvetica* [15], *R. aeschlimannii* [14,17], or *Candidatus* “Rickettsia uralica” [23].

Divided into five biogeographical regions with 21 ecoregions, Romania is a country with high biodiversity [24], including an important variety of tick species and potential tick vertebrate hosts [25,26]. This diversity in both tick species and vertebrates is expected to result in a high diversity of tick-associated pathogens. Despite increasing efforts, the data regarding diversity and distribution of *Rickettsia* species across Romania remain scarce, most studies being either limited to a small sampling area or focused on engorged ticks. To date, several SFG *Rickettsia* have been reported in questing or engorged ticks and less in tissue samples in Romania (Table 1).

Currently, seven out of the eight tickborne rickettsiae with known pathogenicity to humans are present in Europe (except *R. sibirica mongolitimonae*) and also reported in Romania [14] (Table 1), highlighting an increased risk to public health. However, because most SFG *Rickettsia* spp. are detected in ticks collected from different hosts (Table 1), the epidemiology of these pathogens in Romania remains poorly described. Therefore, the aim of the present study was to evaluate the diversity of the SFG *Rickettsia* species, mainly in questing ticks and engorged ticks, in different geographical areas of Romania.

## 2. Materials and Methods

### 2.1. Tick Collection and Identification

A total of 2028 questing and engorged ticks were collected as part of several studies on related topics conducted between March 2010 and May 2020. The selection of samples included in the present study was based on geographical coverage, collection data (the newest samples available), DNA concentration, and tick species. Ticks were identified at the species level by stereomicroscopic examination based on morphological and dichotomous characteristics [40]. After identification, all ticks were preserved individually in 70% ethanol and stored at −20 °C.

A total of 1128 *I. ricinus* ticks included in this study were questing ticks collected by flagging from 183 points in 72 localities in all 41 Romanian counties (Figure 1) as part of a previous study [26]. Ticks from the aforementioned study were first randomly selected to cover the entire territory of Romania. In all selected samples, the DNA concentration was measured to exclude inappropriate samples with low DNA concentration (<30 ng/µL).

Additionally, a total of 672 *Dermacentor* spp. questing ticks collected by flagging were included in this study, namely 507 *D. marginatus* and 165 *D. reticulatus*, all female and male adults. The collection was performed in five counties in northern and northwestern Romania [27].

Similarly, a total of 100 *Hae. punctata* questing ticks were collected by flagging, including larvae, nymphs, and adults. The ticks were collected from six urban and peri-urban sampling sites in Cluj-Napoca forests [28]

Considering the lack of data on SFG *Rickettsia* in *R. rossicus* [41], a tick present in the steppe region of southeastern Romania [42], *R. rossicus* collected from owned dogs (*Canis familiaris*) in three locations in Tulcea county were also included in the study. Overall, 128 *R. rossicus* adults were analysed.

### 2.2. DNA Extraction

DNA extraction was performed using the ISOLATE II genomic DNA kit (Bioline, UK) in compliance with the manufacturer’s instructions. For accuracy and to avoid cross contamination, negative controls were used during each step. DNA from ticks was extracted individually based on species, developmental stage, and location. The concentration and purity of the DNA extract were evaluated in a representative number of samples through a random procedure using a Nanodrop ND-1000 spectrophotometer (NanoDrop Technologies, Inc., Wilmington, DE, USA).

### 2.3. Polymerase Chain Reaction (PCR)

*Rickettsia* spp. DNA amplification protocols were performed using specific primers; initially, primers amplifying a *gltA* gene fragment [43] were used for *Dermacentor* spp. ticks, whereas for the remaining tick species, a nested protocol was carried out, using specific primers amplifying 17-kDa outer-membrane gene fragments [44] (Table 2).

Each reaction mix was performed in a volume of 25 µL consisting of 4 µL DNA isolate, 6.5 µL distilled water, 12.5 µL PCR Master Mix (Rovalab), and 1 µL of each diluted primer (10 pmol/µL). Instead of the DNA isolate, 1 µL PCR product was used for the nested protocol. The analysis was performed using a T1000 thermal cycler (Bio-Rad, Berkeley, CA, USA).

The amplification profile for *gltA* consisted of 5 min of initial denaturation at 95 °C, followed by 35 cycles of denaturation at 95 °C for 30 s, annealing at 53 °C for 30 s, extension at 72 °C for 30 s, and a final extension at 72 °C for 5 min.

For *17-kDa* outer-membrane gene amplification, the profile consisted of an initial denaturation at 95 °C for 3 min, followed by 35 cycles of denaturation at 95 °C for 30 s, annealing at 61 °C for 30 s, elongation at 72 °C for 45 s, and a final elongation at 72 °C for 5 min. The nested step followed the same profile with an annealing temperature of 54 °C.

In each PCR reaction set with 96 samples, 3 positive and 3 negative controls were used, which were randomly placed in the plate to assess the specificity of the reaction and the presence of possible cross contamination. The negative control had the same composition as the mixture to be analysed, with 4 µL PCR water instead of DNA. The positive controls consisted of *Rickettsia* spp. DNA confirmed by sequencing [32].

PCR products were visualized after electrophoresis in 1.5% agarose gel stained with SYBR Safe DNA gel stain (Invitrogen, Waltham, MA, USA) addition. Migration occurred at a continuous current intensity of 110 mA, 100 V, for 30–45 min.

### 2.4. DNA Sequencing

After purifying the PCR products using a FavorPrep™ GEL/PCR purification kit (FAVORGEN-Europe, Wien, Austria), positive amplicons were sequenced at Macrogen Europe BV, Amsterdam, Netherlands. Sequence analysis was performed using Geneious^®^ 9.1.2 software (Auckland, New Zealand) and compared to the sequences present in the GenBank database through BLASTn analysis of the *gltA* and 17-kDa outer-membrane gene fragment sequences (http://blast.ncbi.nlm.nih.gov/Blast.cgi, accessed on 16 May 2022).

### 2.5. Statistical Analysis

Data were statistically analysed using Epi Info 7 software (CDC, Atlanta, GA, USA). The infection prevalence of *Rickettsia* spp., the 95% confidence interval, and infection prevalence based on tick species, developmental stage, sex, and location were analysed using the chi-square independence test. A *p*-value ≤ 0.05 was considered significant.

## 3. Results

A total of 2028 ticks were individually analysed by PCR for the presence of *Rickettsia* spp. The ticks included in the study belonged to five species: *I. ricinus*, *D. marginatus*, *D. reticulatus*, *H. punctata*, and *R. rossicus*. All developmental stages were assessed for *I. ricinus* and *H. punctata*, and only adults were assessed in the case of *Dermacentor* spp. and *R. rossicus* (Table 3).

The overall SFG *Rickettsia* spp. infection prevalence was 5.82% (118/2028; 95% CI: 4.88–6.92). Although the prevalence varied considerably (between 7 and 24.22%; Table 3) between tick species due to uneven sample collection, the prevalence rates were not compared.

The differences in SFG *Rickettsia* spp. prevalence in different developmental stages were analysed for *Hae. punctata* and *I. ricinus*, from which all developmental stages were available for analysis and a similar prevalence was observed for all except *I. ricinus* larvae, in which no *Rickettsia* spp. DNA was found (Table 3).

Infection with *Rickettsia* spp. was detected in 22 of the 41 Romanian counties (53.66%; 95% CI: 37.42–69.34), with a variable prevalence ranging between 1.35% and 50% (Table 4, Figure 1). Differences in prevalence rates among counties were evaluated only in the case of *I. ricinus*. A significant difference between counties was registered for *R. helvetica* (*x*^2^ = 156.87, *d.f.* = 32, *p* < 0.0001) and *R. monacensis* (*x*^2^ = 106.45, *d.f*. = 32, *p* < 0.0001).

According to BLAST analysis, the most prevalent species was *R. monacensis*, with a prevalence of 43.22% (51/118; 95% CI: 34.13–52.66%) of the total positive ticks. The sequences showed a 97.7–100% similarity with various strains (e.g., Acc. No. MF491748, GU827099) isolated in *I. ricinus* from different European countries. *Rickettsia monacensis* was detected in *I. ricinus* (7 females, 11 males, and 12 nymphs), *R. rossicus* (12 females and 2 males), and *Hae. punctata* (1 male, 2 nymphs, and 1 larva) ticks, without a significant difference between tick species, developmental stage, or sex. This species was detected in ticks in 16 counties (Figure 1).

The second most prevalent species detected was *R. helvetica*, representing 38.14% (45/118, 95% CI: 29.35–47.53) of the positive ticks, with a similarity of 99–100% with various strains (e.g., Acc. No. KY319214, GU827035) isolated from *I. ricinus* from different European countries. *Rickettsia helvetica* was detected in *I. ricinus* (12 females, 13 males, and 10 nymphs) and *R. rossicus* ticks (6 females and 4 males), without any significant differences. This species was detected in ticks originating from 15 counties (Figure 1).

*Rickettsia raoultii* was identified in 15.25% of the positive ticks (18/118, 95% CI: 9.3–23.03%), with a 99–100% similarity to a strain (Acc. No. JX683120) isolated from *D. marginatus* in Romania. *Rickettsia raoultii* was detected in *D. marginatus* (4 females and 4 males), *I. ricinus* (2 males and 3 nymphs), *R. rossicus* (2 females and 1 male), and *Hae. punctata* ticks (1 male, nymph, and larva), without a significant difference. This species was detected in ticks originating from nine counties (Figure 1).

*Rickettsia slovaca* had a 1.69% prevalence (2/118, 95% CI: 0.21–5.99%), with a similarity of 99.7–100% with a strain (Acc. No. JX683122) isolated from *D. marginatus* in Romania. It was detected only in *D. marginatus* ticks (1 female and 1 male), without a significant difference. This species was detected in ticks originating from two counties (Figure 1).

The sequence isolated from one *R. rossicus* collected from a dog representing 0.85% (1/118, 95% CI: 0.02–4.63%) of positive ticks showed a 99% similarity with *R. hoogstraalii*, various strains (Acc. No. FJ767736, MT010837, MH383145, MN1501180) isolated from *Hae. sulcata* in Croatia, *Argas transgariepinus* in Namibia and South Africa, and *Amblyomma transversale* in the United Arab Emirates.

In one case, the sequence isolated from *I. ricinus* presented low quality and remained unidentified.

## 4. Discussion

Although several studies have been published, data regarding the presence of *Rickettsia* spp. and their epidemiological situation in Romania remain poorly defined. In addition, the geographical distribution and species diversity of ticks in different countries are continuously changing [14,45,46]. To the best of our knowledge, this study represents the first detection of SFG *Rickettsia* spp. in *R. rossicus* adults and *Hae. punctata* larvae and the first report of *R. hoogstraalii* in Romania.

The overall *Rickettsia* spp. prevalence detected in the present study was around 5%, similar to the results reported in other studies in Europe [14,17]. The *Rickettsia* spp. prevalence was similar in adult and immature stages. In the present study, *R. monacensis* and *R. raoultii* were detected in *Hae. punctata* larvae, which highlights the possible transovarial transmission of these pathogens. Similarly, *R. monacensis* and *R. helvetica* have been previously detected in *I. ricinus* larvae [14,47,48].

A difference in *Rickettsia* spp. prevalence between sexes was observed only in *R. rossicus* ticks collected from hosts (33% in females vs. 13.5% in males). This difference may be explained by the different feeding behaviour, with females feeding on a larger volume of blood and for a longer period compared to males [49]. However, in this case, the same difference would be expected for the other Ixodidae species.

*Rickettsia* spp. were detected in more than half of the counties, showing a wide distribution of these pathogens across Romania. Five counties registered a high prevalence (≥20%). However, the number and diversity of collected ticks varied among the counties, making an appropriate interpretation of the results difficult. The importance of the wide distribution and high prevalence is driven by the zoonotic potential of most SFG *Rickettsia* detected to date in Europe [14,46]

In the present study, five different SFG *Rickettsia* species were identified, some of which were associated with *R. rossicus* for the first time. Among these, four species (i.e., *R. helvetica*, *R. monacensis*, *R. slovaca*, and *R. raoultii*) are recognised as human pathogens, and three (*R. helvetica*, *R. monacensis*, and *R. raoultii*) can be transmitted by *I. ricinus* [14,46], which is also the most common tick feeding on humans in Romania [50].

Although *I. ricinus* and *D. marginatus* are well-known vectors of SFG *Rickettsia* [14,17], in the present study, the tick species that displayed the highest diversity of *Rickettsia* was *R. rossicus*, followed by *Hae. punctata*. However, in the case of *R. rossicus*, the DNA origin could be either the blood meal (*R. rossicus* were collected from dogs) or the tick. However, the high prevalence obtained in *R. rossicus*, with the majority of positive ticks collected from different individuals, does not support the blood origin. In addition, dogs are known host of only a few SFG *Rickettsia*, such as *R. conorii* [14,17,51], *R. rickettsii* [52], *R. parkeri* [53], and *R. japonica* [54].

The most often detected species in the present study was *R. monacensis*. To the best of our knowledge, this study represents its first detection in *R. rossicus* ticks and *Hae. punctata* larvae. Considered one of the most common species of *Rickettsia* in Europe [55], *R. monacensis* was detected in Turkey [56], Spain [57], the Netherlands [58], and Serbia [59], as well as and with a lower prevalence in Iceland, Russia, Italy, and Sweden [60,61,62,63]. This species is transmitted by *I. ricinus*, and it was isolated from humans and lizards, which were suggested as reservoir hosts [14,17]. The previous detection of *R. monacensis* in different tick species collected from dogs, including *R. rossicus* in the present study, suggest either the implication of other species as possible vectors [64] or the possible infection of dogs with this rickettsia.

The second most prevalent rickettsia species was *R. helvetica*, detected in *I. ricinus* and *R. rossicus.* Similar to *R. monacensis*, the main vector is *I. ricinus* [65]. This *Rickettsia* species is common in Europe, being detected over most of its territory [14]. Concerns about this species are substantiated by the impact it has on human health, as described in several studies [66,67,68,69].

Other species identified in our study, such as *R. raoultii* and *R. slovaca*, are also important human pathogens, being the causative agents of SENLAT (scalp eschar and neck lymphadenopathy), with implications for both animal and human health [70]. In Romania, SENLAT was detected in several patients [37]. The most frequently involved tick species in transmitting infection is *D. marginatus* [71], followed by *D. reticulatus* [72]. The presence of SENLAT agents was also reported in *Hae. inermis*, *Hae. bispinosa* [73], and *I. ricinus* [74]. This pathogen was reported in Spain [75], Hungary [76], France [71], Italy [77], Bulgaria [78], and Poland [79].

In one case, the sequence analysis of a *Rickettsia* spp. isolated from one engorged male of *R. rossicus* was identified as *R. hoogstraalii. Rickettsia hoogstraalii* was isolated for the first time in 2006 in *Hae. sulcata* ticks collected from sheep and goats in Croatia [80] as well as *Carios capensis* ticks from the United States [81]. Since then, it has been detected in hard ticks collected from domestic and wild ruminants across Europe, in Spain (in *Hae. Punctata* and *Hae. sulcata*), Cyprus (*Hae. punctata*), Italy, Sardinia (*Hae. Punctata* and *Hae. sulcata*), Greece (*Hae. parva* and *Hae. sulcata*), Turkey (*Hae. parva*), and Georgia (*Hae. sulcata* and *D. marginatus*) [57,82,83,84,85,86]. It has been also detected in other tick species, mainly Argasidae, in different parts of the world, including Japan [87], Ethiopia [88] the western Indian Ocean islands [89], Iran [90], Namibia [91], Zambia [92], China [93], and the UAE [94]. The pathogenic potential of *Rickettsia hoogstraalii* is poorly understood and is considered similar to *R. felis* [95]. In addition, it was reported to cause a cytopathic effect in Vero, CCE3, and ISE6 cells [80].

This study represents both the first detection of *R. hoogstraalii* in *R. rossicus* and the first detection of this Rickettsia species in Romania. Nevertheless, the obtained short *17-kDa* outer membrane gene sequence does not allow for a clear conclusion with respect to this detection. In addition, further studies should be conducted to evaluate the pathogenicity of *R. hoogstraalii* for mammals, as well as the involvement of *R. rossicus* as a possible vector species.

## 5. Conclusions

In this study, we reported a wide distribution of SFG *Rickettsia* across Romania, including well-known human pathogens, such as *R. helvetica*, *R. raoultii*, and *R. slovaca*, or possible zoonotic pathogens, such as *R. monacensis* and *R. hoogstraalii*, raising concerns about the risks posed to public health.

The detection of *R. monacensis* and *R. raoultii* in *Hae. punctata* questing larvae strongly suggests the transovarial transmission of these pathogens and supports the possible involvement of this tick species as a vector.

The detection of *R. hoogstraalii*, *R. helvetica*, *R. monacensis*, and *R. raoultii* in *R. rossicus* ticks collected from dogs suggests either the possible involvement of this tick species as a vector for multiple SFG *Rickettsia* or the possible infection of dogs with these species.

Further studies are required to confirm the presence of *R. hoogstraalii* in Romania. Moreover, the transovarial transmission of SFG *Rickettsia* in *Hae. punctata* ticks and the vectorial implication of *R. rossicus* or dogs as reservoir hosts for multiple SFG *Rickettsia* species require additional research to be confirmed.

## Figures and Tables

**Figure 1 vetsci-09-00343-f001:**
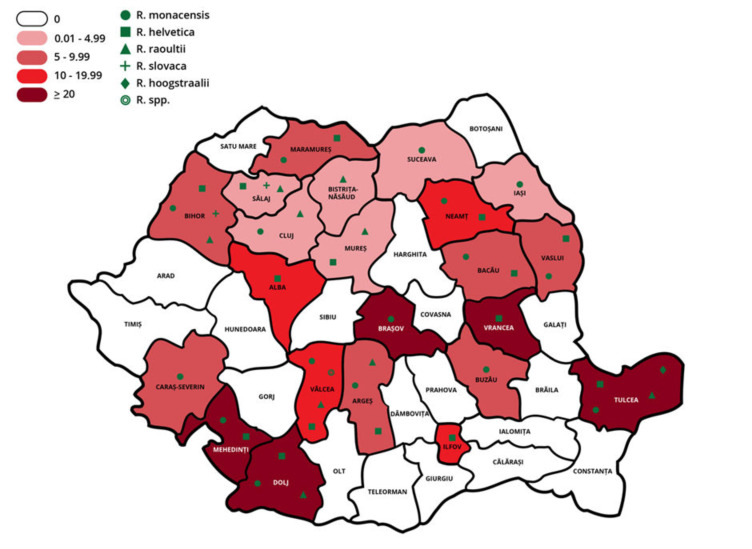
The prevalence, diversity, and geographic distribution of *Rickettsia* spp. in ticks in Romania.

**Table 1 vetsci-09-00343-t001:** Overview of Rickettsia species in questing and engorged ticks, as well as vertebrate host tissues, in Romania.

*Rickettsia* spp.	Tick Species	Host Species	County	Reference
Questing ticks				
*Rickettsia* spp. ^1^	*D. marginatus, D. reticulatus*	-	Cluj	[27,28]
*R. helvetica* ^1^	*I. ricinus, Hae. punctata*	-	Cluj	[28]
	*I. ricinus*	-	Iași, Tulcea	[29]
*R. monacensis* ^1^	*I. ricinus*	-	Iași, Tulcea	[29]
	*I. ricinus, Hae. punctata*	-	Cluj	[28]
*R. raoultii* ^1^	*D. reticulatus*	-	Iași	[29]
*R. conorii* ^1^	*Hae. punctata*	-	Cluj	[28]
Ticks collected from hosts				
*R. helvetica* ^1^	*I. ricinus*	*Bos taurus, Equus caballus*	Ilfov, Prahova	[30]
	*I. ricinus, I. arboricola, I. redikorzevi*	*Erithacus rubecula, Panurus biarmicus, Turdus merula, T. philomelos*	Constanța	[31]
	*I. ricinus*	*Homo sapiens*	Cluj	[32]
	*I. ricinus*	*H. sapiens*	Sibiu	[33]
	*I. ricinus, I. hexagonus*	*Erinaceus roumanicus*	Cluj	[28]
	*I. ricinus*	*E. rubecula, T. merula*	Cluj	[28]
	*I. ricinus*	*Talpa europea*	Cluj	[28]
*R. monacensis* ^1^	*I. ricinus*	*Canis familiaris*	Ilfov	[34]
	*I. ricinus, I. arboricola, Hae. concinna*	*E. rubecula, T. merula, T. philomelos*	Constanța	[31]
	*I. ricinus*	*H. sapiens*	Cluj	[32]
	*I. ricinus*	*H. sapiens*	Sibiu	[33]
	*I. ricinus, D. reticulatus, R. sanguineus*	*C. familiaris, Felis catus, Ovis aries, Vulpes vulpes*	Satu-Mare, Călărași, Ilfov, Timiș, Dâmbovița, Mehedinți	[35]
	*I. ricinus, I. hexagonus, Hae. punctata*	*E. roumanicus*	Cluj	[28]
	*Hae. concina*	*Sturnus vulgaris*	Cluj	[28]
*R. raoultii* ^1^	*D. marginatus*	*B. taurus, O. aries*	Dâmbovița, Satu-Mare, Vâlcea	[30]
	*D. reticulatus*	*C. familiaris*	Ilfov	[34]
	*D. marginatus*	*H. sapiens*	Sibiu	[33]
	*D. marginatus, D. reticulatus, R. sanguineus*	*C. familiaris, Capra hircus, O. aries*	Ilfov, Călărași, Covasna, Dâmbovița, Bistrița-Năsăud, Mehedinți, Vâlcea	[35]
*R. slovaca* ^1^	*D. marginatus*	*B. taurus*	Dâmbovița	[30]
	*D. reticulatus*	*C. familiaris*	Ilfov	[34]
	*I. ricinus*	*T. merula*	Constanța	[31]
	*I. ricinus, D. marginatus, R. sanguineus*	*C. familiaris, Capra hircus, O. aries, V. vulpes*	Călărași, lfov, Covasna, Timiș, Mehedinți, Vâlcea	[35]
*R. aeschlimannii* ^1^	*Hy. marginatum*	*B. taurus*	Bistrița-Năsăud	[35]
	*Hae. concina*	*S. vulgaris*	Cluj	[28]
*R. conorii* ^1^	*R. sanguineus*	*C. familiaris*	Ilfov	[34]
*R. felis* ^1^	*I. ricinus*	*T. merula*	Cluj	[28]
*R. massiliae* ^1^	*I. ricinus, I. arboricola*	*T. philomelos*	Constanța	[31]
Vertebrate host tissues				
*R. helvetica* ^1^	- -	*Parus major, Corvus frugilegus*	Cluj	[28]
		*E. roumanicus*	Cluj	[28]
*R. monacensis* ^1^	-	*T. merula*	Cluj	[28]
	-	*Apodemus agrarius, A. sylvaticus, Mus musculus*	Cluj	[28]
	-	*Pipistrellus pipistrellus, Nyctalus noctula*	Alba, Neamț	[36]
*R. slovaca* ^2^	-	*H. sapiens*	Ilfov	[37]
*R. massiliae* ^2^	-	*H. sapiens*	Ilfov	[37]
*R. slovaca*/*R. raoultii* ^2^	-	*H. sapiens*	Ilfov	[37]
*R. conorii* ^3^	-	*H. sapiens*	Unspecified	[38]
*R. conorii* ^3^	-	*H. sapiens*	Unspecified	[39]

^1^ Molecular detection, ^2^ Western blot, ^3^ Immunofluorescence.

**Table 2 vetsci-09-00343-t002:** Primers used for the detection of Rickettsiales DNA in ticks.

Fragments of Genes	Names of Gene	Citations
Rsfg877: GGGGGCCTGCTCACGGCGG	*gltA*	[43]
Rsfg1258: ATTGCAAAAAGTACAGTGAACA		
rickP3: GGAACACTTCTTGGCGGTG	17-kDa	[44]
rickP2: CATTGTCCGTCAGGTTGGCG		
rickP5: GCATTACTTGGTTCTCAATTCGG		
rickP4: AACCGTAATTGCCGTTATCCGG		

**Table 3 vetsci-09-00343-t003:** *Rickettsia* spp. prevalence and its distribution according to tick species, developmental stage, and sex.

*Tick* sp.	Origin	Developmental Stage	Sex	Prevalence % (*n*/Total)	95% CI
*Haemaphysalis punctata*	Questing	AD	F	7.14% (1/14)	0.18–33.87
			M	7.14% (1/14)	0.18–33.87
		N		6.98% (3/43)	1.46–19.06
		L		6.9% (2/29)	3.99–10.04
Total				7.00% (7/100)	5.02–7.87
*Ixodes ricinus*	Questing	AD	M	6.84% (27/395)	4.74–9.76
			F	6.55% (19/290)	3.99–10.04
		N		5.83% (25/429)	3.98–8.46
		L		0% (0/14)	NA
Total				6.29% (71/1128)	2.86–13.89%
*Dermacentor marginatus*	Questing	AD	F	1.98% (5/253)	0.64–4.55%
			M	1.57% (4/254)	0.43–3.98%
Total				1.78% (9/507)	0.94–3.34%
*Dermacentor reticulatus*	Questing	AD	F	0% (0/94)	NA
			M	0% (0/71)	NA
Total				0% (0/128)	NA
*Rhipicephalus rossicus*	Engorged	AD	F	33.33% (23/69)	22.44–45.71%
			M	13.56% (8/59)	6.04–24.98%
Total				24.22% (31/128)	17.09–32.58%

*n*/total: positive ticks/total ticks; CI: confidence interval; AD: adult; F: female; M: male; N: nymph; L: larva.

**Table 4 vetsci-09-00343-t004:** *Rickettsia* spp. prevalence in Romanian counties.

County	Prevalence % (*n*/Total)	95% CI
Alba	18.18 (2/11)	2.28–51.78
Argeș	6 (6/100)	2.23–12.60
Bacău	5.63 (4/71)	1.56–13.8
Bihor	6.1 (10/164)	2.96–10.93
Bistrița-Năsăud	2.50 (1/40)	0.06–13.16
Brașov	20 (1/5)	0.51–71.64
Buzău	7.5 (3/40)	1.57–20.39
Cluj	3.5 (7/200)	1.42–7.08
Covasna	7.5 (3/40)	1.57–20.39
Dolj	50 (5/10)	18.71–81.29
Ilfov	10 (2/20)	1.23–31.70
Iași	4.12 (4/97)	1.13–10.22
Mehedinți	30 (3/10)	6.67–65.25
Maramureș	8 (6/75)	2.99–16.6
Mureș	3.64 (4/110)	1.00–9.05
Neamț	10 (5/50)	3.33–21.81
Sălaj	1.35 (4/297)	0.37–3.41
Suceava	1.67 (1/60)	0.04–8.94
Tulcea	23.87 (37/155)	17.4–31.37
Vâlcea	15 (4/40)	5.71–29.84
Vrancea	20 (2/10)	2.52–55.61
Vaslui	9.09 (2/22)	1.12–29.16

## Data Availability

Not applicable.

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
