# Peer review of "Spotted Fever Group Rickettsia spp. Diversity in Ticks and the First Report of Rickettsia hoogstraalii in Romania"

_vetsci, 2022, doi:10.3390/vetsci9070343_

Round 1

Reviewer 1 Report

Dear authors,

The manuscript was very interesting. I recommend some changes.

L 2: SFG should be the full name

L78: *and ** from Western Blot and Immunofluorescence, respectively should be changed, as it is used for the footnote and it is confusing the reader

L 157, 186, 196, 202, 206: remove the word “statistically”

L 269: “beingt he” should be “being the”

Author Response

Dear reviewer 1:

Thank you for your valuable comments and suggestions. Below you can find our response point by point. The changes are mark in manuscript with yellow.

L 2: SFG should be the full name

R: done

L78: *and ** from Western Blot and Immunofluorescence, respectively should be changed, as it is used for the footnote and it is confusing the reader

R: were replaced both in the table and footnote as: 1Molecular detection, 2Western Blot, 3Immunofluorescence

L 157, 186, 196, 202, 206: remove the word “statistically”

R: done

L 269: “beingt he” should be “being the”

R: done

Reviewer 2 Report

All comments and sugestions are in the enclosed file

Author Response

Dear reviewer 2:

Thank you for your valuable comments and suggestions. Below you can find our response point by point. Changes in the manuscript are marked with yellow.

Romania.

R: accepted

Order the values in descending order

R: done

e.g. I. ricinus

R: replaced with e.g. Ixodes ricinus

Table 1

R: Title: Table 1. Overview of Rickettsia species in questing and engorged ticks and vertebrate host tissues in Romania.

Table head: replaced with Tick species and Host species

Changed H. sapiens in Homo sapiens

At the suggestion of the reviewer 1 the table footnote and first column were changed to clearly indicate the detection method: 1Molecular detection, 2Western Blot, 3Immunofluorescence

please enter a number of examined I.ricinus

R: added

please specify the values (low DNA concentration)

R: added

Add were, delete were also included

R: done

Please make up the table with names of the primers, which were used in the study (with fragments of genes, names of genes and citations)

R: done

Why the different type of primers were used for Dermacentor spp, and the another type for the remaining tick species ??

Look at Portillo et al. Guidelines for the detection of Rickettsia spp. 2017

R: we initially used gltA as we already used this protocol in previous studies (Matei et al., 2017; 2021). However due to the low prevalence observed in Dermacentor we decided to continue with a nested protocol, which has in general a higher sensibility.

please enter the full names of species (table 2)

R: done

It is very important information

R: It was mentioned as a limitation of the study. The sequence analysis allowed a clear identification of the species. However, a deeper phylogenetic analysis requires several markers and longer sequences.